# Ethanol Hormesis in Honeybees (*Apis mellifera* L.) Infected with *Vairimorpha (Nosema*) spp.

**DOI:** 10.3390/ani15223316

**Published:** 2025-11-17

**Authors:** Karolina Kuszewska

**Affiliations:** Department of Zoology and Animal Welfare, Faculty of Animal Science, University of Agriculture in Krakow, Al. Mickiewicza 21, 31-120 Krakow, Poland; karolina.kuszewska@urk.edu.pl

**Keywords:** *Apis mellifera*, honeybee, hormesis, *Nosema* sp., *Vairimorpha* spp., ethanol, alcohol

## Abstract

This study examined honeybees to test ethanol hormesis in the context of *Vairimorpha (Nosema)* spp. infection. It explored whether small amounts of ethanol could mitigate infection effects and influence lifespan. The results showed a biphasic response: low to moderate ethanol exposure extended lifespan and aligned with reduced *Vairimorpha* spp. severity in infected bees, consistent with hormesis. In contrast, higher ethanol exposure was toxic, increasing mortality and parasite burden. Overall, the results suggest that trace ethanol exposure—similar to what bees may encounter in nectar—can differentially affect bee health based on infection status, highlighting nuanced associations among diet, infection, and longevity.

## 1. Introduction

Hormesis is a biological phenomenon defined by a biphasic response to low doses of potentially harmful agents, such as toxins or radiation, which can produce beneficial effects [1,2,3]. While high doses are often harmful, low doses can promote health, enhance growth, and improve resilience [1]. This paradox has attracted significant interest across toxicology, pharmacology, and environmental science [4,5]. The concept of hormesis emerged in the early 20th century when researchers observed that certain toxic substances could have stimulatory effects at lower concentrations. For example, low levels of radiation exposure were found to enhance cellular repair mechanisms, leading to improved survival rates and reduced cancer incidences, challenging the traditional “linear no-threshold” model of toxicity [4,6].

Hormesis is evident in plants, which activate defense mechanisms when exposed to mild stressors like drought or pathogens, improving their resilience [7,8]. This phenomenon is also observed in animals and humans [3,9]. In vertebrates, low doses of heavy metals, such as arsenic and lead, can stimulate antioxidant defenses, enhancing survival against higher toxin concentrations [10,11]. Research in mammals, particularly rodents, shows that low doses of ionizing radiation can reduce cancer rates through activated DNA repair mechanisms, promoting cellular resilience and potentially longevity [12,13]. Invertebrates, which represent the majority of animal diversity, also exhibit hormetic responses. Marine invertebrates, like mollusks and crustaceans, demonstrate increased resilience when exposed to low levels of pollutants, such as cadmium, resulting in improved survival and reproductive success [14,15]. Insects, including fruit flies (*Drosophila melanogaster*), show enhanced stress resistance and lifespan when exposed to low pesticide doses [16,17].

Hormesis is also observed in various pollinator species, where low doses of harmful substances can lead to beneficial effects. For instance, adult female *Osmia* bees, which are solitary bees, demonstrated increased longevity when exposed to sublethal doses of pesticides such as Glyphosate (a phosphonate herbicide) and Clothianidin (a neonicotinoid insecticide) [18,19]. In honeybees, chronic exposure to imidacloprid (neonicotinoid insecticide) resulted in a biphasic response to varying concentrations, leading to opposite effects on body weight and gene expression, as well as a dose-dependent impact on flight ability [20]. Interestingly, some of these benefits may come at a cost later in life. For example, while neonicotinoids enhanced certain traits in female *Osmia*, they also negatively affected the reproduction of their offspring, even if these offspring were raised in a pesticide-free environment [21]. This illustrates the complex and lasting effects of pesticides on pollinators, revealing both stimulatory and harmful impacts. Another example of hormesis comes from honeybees, where substances like caffeine and nicotine, typically toxic at higher concentrations [22,23], have been shown to extend survival by 57% when administered in low doses [24]. Moreover, honeybee workers demonstrated improved learning and memory abilities [25], while ants exhibited enhanced foraging skills after exposure to nicotine and caffeine [26]. Additionally, injecting sublethal doses of nicotine into the antennal lobes of honeybee workers increased their sucrose sensitivity and olfactory learning [27]. While it remains unclear whether these various examples can be consolidated into a single mechanism, it is evident that hormetic effects can vary significantly depending on the specific trait being examined, as well as the age and life stage of the individual [28]. Overall, the exploration of hormesis in pollinators underscores the intricate balance between stress responses and resilience in the face of environmental challenges.

Another substance that negatively impacts the vital functions of insects and may exhibit the phenomenon of hormesis is ethanol, which is commonly found in flower nectar and fruit as a results of yeast fermentation [29,30,31,32]. Hormetic responses to ethanol and its metabolite acetaldehyde have been documented in insects such as fruit flies, indicating adaptive tolerance to ethanol-rich environments [33]. The factors and settings that promote hormesis warrant further study, especially in agroecosystems like orchards, where ripening fruit accumulate ethanol. Some orchard yeasts tolerate ethanol up to 20%, suggesting high exposure is possible in these habitats [34]. In general, ethanol consumption can have significant adverse effects on honeybees, particularly when they are exposed to high concentrations [35,36]. One of the most immediate negative impacts of ethanol on honeybees is impaired motor skills [37]. When bees consume excessive amounts of ethanol, their coordination, ability to fly, and cognitive functions are severely affected [38,39,40,41]. This disorientation can hinder their ability to navigate back to their hives, leaving them vulnerable to predators and environmental hazards. As a result, this disorientation not only increases mortality rates but also disrupts foraging efficiency, leading to a decrease in food collection for the entire colony [40]. Despite the negative effects of ethanol on bees, some studies suggest that bees may prefer sugar solutions with a small amount of ethanol added over pure sugar solutions [42,43,44]. This behavior suggests that low concentrations of ethanol might offer some benefits to bees. One such benefit could be a reduction in the levels of parasites commonly found in bees, such as the microsporidium *Vairimorpha (Nosema)* spp., which causes nosemosis—a widespread bee disease. *Vairimorpha (Nosema)* spp. (formerly known as *Nosema* spp.—*Nosema apis* and *Nosema ceranae*). Taxonomic revisions prompted by molecular and phylogenetic analyses have reassigned these species to genera *Vairimorpha* [45,46]. The microsporidia responsible for this condition, historically referred to as “nosemosis,” impose energetic stress on honeybee hosts and produce highly resistant spores capable of persisting for years. *Vairimorpha ceranae* was first identified as an infective agent for *Apis cerana* [47], and was subsequently detected in *Apis mellifera* [48] with its spread linked to broader disease dynamics and, in some regions, colony declines (e.g., [45,46,47,48,49,50,51,52]).

In honeybees, *Vairimorpha* spp. infections have been major research targets due to their economic and ecological impacts, and the pathogens are known to impair host energy metabolism [53,54,55]. The spores of *Vairimorpha* spp. are extremely resistant to external stress factors and can survive for several years without losing their ability to infect other insects [50]. For this reason, combating nosemosis is quite challenging. There are reports that some beekeepers add ethanol to sucrose solutions before winter to prevent *Vairimorpha* spp. infections and to treat already infested colonies [43]. However, studies conducted so far do not confirm a positive relationship between ethanol and the treatment of *Vairimorpha* spp. This may be related to the high doses of ethanol chosen by researchers, as even the lowest dose of 2.5% can have a negative effect on bee survival [43].

The goal of this study was to assess the effect of ethanol on bees affected by nosemosis. We aimed to investigate the phenomenon of ethanol hormesis in individual honeybees infected with *Vairimorpha* spp. spores. To conduct this experiment, we collected foraging bees from the hive and divided them into fourteen groups, each consisting of 50 individuals. Seven groups contained bees infected with *Vairimorpha* spp. spores, while the remaining seven groups were uninfected. Both infected and uninfected bees were exposed to different ethanol concentrations (0%, 0.0313%, 0.625%, 1.25%, 2.5%, 5%, and 10%). We hypothesized that bees exposed to low ethanol concentrations (below 2.5%) would be less infected by the parasite and would live longer than those exposed to higher ethanol concentrations (2.5% and above) or those with no access to ethanol after being infected with *Vairimorpha* spp.

## 2. Methods

The study was conducted in May and June 2023 at the experimental apiary of the University of Agriculture in Krakow (Krakow, southern Poland), involving four unrelated, queen-right honeybee (*Apis mellifera carnica*) colonies, each consisting of 20,000 to 40,000 workers and led by naturally mated queens. All colonies were treated in the same manner, with procedures initiated every 10 days, systematically addressing each colony in turn. Initially, forager bees were separated from non-foraging bees. To achieve this, each colony was divided into two subunits (A and B) at 09:00 h, prior to the active foraging period (see Figure 1). Subunit A, which included all the worker bees and several frames with food, was moved a few meters away from the original colony site. Subunit B, containing a queen and all frames with brood of various ages but no adult workers, remained at the original location, with its entrance aligned in the same direction as that of the native hive. This manipulation ensured that only nurse bees were present in subunit A, as all foragers returned from the field to subunit B after departing from subunit A (Figure 1b) [56]. On the same day, individuals were captured from subunit B and randomly divided into fourteen groups of 50 individuals each (Figure 1c).

Bees from the first seven groups (1–7) were individually infected with Nosema spp. spores, while bees from the remaining seven groups (8–14) were not infected. Specifically, each adult bee in groups 1–7 was placed in a Petri dish with 10 µL of a 50% water and sugar solution containing 1.75 × 10^4^
*Vairimorpha* spp. spores. Bees in groups 8–14 were treated similarly but were fed only 20 µL of the water and sugar solution without *Vairimorpha* spp. spores. This method ensured that each infected worker received a consistent number of parasite spores. The bees in these fourteen groups were housed in wooden-frame cages (13 × 9 cm with a height of 5 cm, sides made of glass and steel mesh) and provided with a small piece of bee comb. Each cage represented one experimental group, containing 50 bees. The cages were incubated at 36 °C with 50–60% relative humidity, and a 50% sucrose solution along with water was available ad libitum.

Starting the next day, cages containing both uninfected and *Vairimorpha* spp.-infected bees were divided into seven groups and exposed to daily ethanol (EtOH) solutions at the following concentrations: 0%, 0.0313%, 0.625%, 1.25%, 2.5%, 5%, and 10% (respectively, cages 1–7 correspond to the uninfected group and cages 8–14 to the infected group; Figure 1c). Bees from these cages were fed individually once daily with sugar–water syrup (1:1) supplemented with ethanol at the appropriate concentration, or with only the sugar–water solution. In this feeding step, similar to the *Vairimorpha* spp. spore feeding, each bee was placed in a Petri dish and given 20 µL of a 50% sugar solution in water, with the appropriate concentration of ethanol (Figure 1c). This feeding manipulation was repeated from days 2 to 9 of the experiment. After 9 days (on day 10), the bees were immediately frozen and subsequently dissected. Additionally, during this period (days 1–9), the cages were checked daily, and any dead bees were counted and removed.

The frozen bees were counted, and the number of *Vairimorpha* spp. spores in the digestive tract was also quantified. For this, the digestive tract (excluding the crop) of each bee was homogenized in 300 μL of distilled water. The *Vairimorpha* spp. spores were counted using a Bürker haemocytometer in a total solution volume of 1.25 × 10^−2^ μL. If the number of spores counted per sample was fewer than 10, the total solution volume per sample was increased to 8 × 10^−2^ μL. The total number of spores per bee was determined using the following formula: the number of spores per bee = (the number of spores per sample × 300 μL)/(the total solution volume of the sample) [57].

We estimated differences in survival curves using a log-rank test. Initially, we examined whether there were differences in survival curves among bees from four different colonies. Since no significant differences were observed between the experimental colonies, we combined the data from these colonies and compared the survival curves of the experimental groups. A post hoc pairwise comparison was conducted to identify groups that differed from one another [58]. The levels of infection were analyzed using a mixed-model three-way ANOVA, with the *Vairimorpha* spp. infection and ethanol concentration as a fixed effect and the colony of origin as a random effect. Statistically significant ANOVA results were followed by multiple comparisons using the post hoc Tukey HSD test, with a *p*-value of <0.05 considered significant.

## 3. Results

One of the most crucial aspects of ensuring that the results were meaningful was verifying that the foraging bees collected from the colonies were free from *Vairimorpha* spp. This was vital because prior infections could influence lifespan and other significant anatomical parameters, including the number of tested *Vairimorpha* spp. spores. This verification involved conducting a three-way ANOVA, with *Vairimorpha* spp. spores feeding status (fed—1 vs. not fed—0) as a fixed factor, ethanol exposure as another fixed factor, and the colony from which the workers originated as a random factor. The results showed statistically significant differences between workers from groups fed with *Vairimorpha* spp. and those not fed with *Nosema* sp. (three-way ANOVA: F = 11.54, df = 2, *p* < 0.001; see Table 1 and Appendix A). Bees not fed with *Vairimorpha* spp. had no spores of this microsporidium in their gut. Moreover, statistically significant differences were also found between workers exposed to different ethanol concentrations (three-way ANOVA: F = 529.0, df = 6, *p* < 0.001; see Table 1, Figure 1, and Appendix A). Additionally, the interaction between these two factors (and *Vairimorpha* spp. feeding and ethanol exposure) was statistically significant (three-way ANOVA: F = 529.0, df = 6, *p* < 0.001). On the other hand, the colony from which the bees originated (three-way ANOVA: F = 1.0, df = 3, *p* = 0.507), as well as interactions between colony and *Vairimorpha* spp. feeding (F = 1.2, df = 3, *p* = 0.312), colony and ethanol concentration (F = 1.0, df = 18, *p* = 0.500), and colony, *Vairimorpha* spp. feeding, and ethanol concentration (F = 1.3, df = 18, *p* = 0.189), were not statistically significant. Post hoc comparisons indicated that the highest number of *Vairimorpha* spp. spores was found in workers from groups 2, 3, and 6, which were infected with *Vairimorpha* spp. and exposed to ethanol concentrations of 0%, 0.0313%, and 2.5%, respectively. Bees from group infected with *Vairimorpha* spp. and exposed to 5% ethanol had the next highest spore counts. Following them were bees from groups infected with *Vairimorpha* spp. and exposed to 0.625% and 1.25% ethanol. The lowest number of *Vairimorpha* spp. spores among the infected groups was observed in bees from groups exposed to 10% ethanol. The groups that were not exposed to *Vairimorpha* spp. spores showed no infection. All results are presented in Table 1 and Appendix A, while the results for workers from the *Vairimorpha* spp. fed groups are illustrated in Figure 2.

Next, to determine the mortality of workers exposed to different concentrations of ethanol, the lifespan of caged workers was analyzed from the time they were placed in the cages (day 0) until the 10th day of the experiment, when the remaining workers were frozen to estimate the number of *Vairimorpha* spp. spores in their digestive tracts. Initially, survival curves from different colonies but within the same experimental group were analyzed to see if the colonies differed from one another. The results showed that there were no significant differences between workers from different colonies and the results you can see in Table 2.

Therefore, in the subsequent analysis, where the lifespan between groups was examined, data from different colonies were combined. The results indicated that workers fed with *Vairimorpha* spp. had a shorter lifespan than those that were not fed *Vairimorpha* spp. (log-rank test χ^2^ = 10.112, N = 2800, censored = 1608, uncensored = 1192, df = 2, *p* < 0.001). There were also differences in survival between individuals exposed to different ethanol concentrations (log-rank test χ^2^ = 756.902, N = 2800, censored = 1608, uncensored = 1192, df = 6, *p* < 0.001).

Analyses comparing the lifespan of individuals exposed to different ethanol concentrations showed that the effects of ethanol dose differ depending on whether bees were fed *Vairimorpha* spp. spores or not. The individuals not fed with *Vairimorpha* spp. spores demonstrated significant differences between the groups (log-rank test χ^2^ = 405.809, N = 1400, censored = 947, uncensored = 453, df = 6, *p* < 0.001). Post hoc tests revealed that only ethanol concentrations of 2.5% or higher exert a toxic effect on bees, with toxicity increasing at higher doses. Conversely, low ethanol doses of 0.0313%, 0.0625%, and 1.25% did not impact lifespan, and bees in these groups lived as long as the control group, which received no ethanol at all (0%) (see Figure 3A, Table 3, and Appendix A). However, for individuals fed with *Vairimorpha* spp. spores, the overall results also showed differences in their lifespan (log-rank test χ^2^ = 415.959, N = 1400, censored = 661, uncensored = 739, df = 6, *p* < 0.001). In contrast, post hoc tests showed that, unlike the non-fed individuals, the longest-lived groups were those exposed to 0.625% and 1.25% ethanol. These were followed by bees with no ethanol exposure (0%), as well as those exposed to 0.0313% and 2.5% ethanol. Shorter lifespans were observed in individuals exposed to 5% ethanol, with the shortest lifespans seen in those exposed to 10% ethanol (see Figure 3B, Table 3, and Appendix A).

## 4. Discussion

In our study, we focused on honeybees, specifically investigating whether low concentrations of ethanol could alleviate the harmful effects of *Vairimorpha* spp. infection. The implications of ethanol on honeybee health are significant, given its presence in natural floral nectars [29,31]. Previous research has shown that high ethanol concentrations can severely impair honeybee motor skills [37], cognitive functions, and coordination, leading to disorientation [38,39,40,41] and increased mortality rates [35]. Our findings support these observations; among uninfected bees, ethanol exposure also exhibited varied effects depending on the concentration. Bees exposed to low concentrations of ethanol (0%, 0.0313%, 0.625%, and 1.25%) had lifespans comparable to those in the control group [59], whereas exposure to concentrations of 2.5% and higher resulted in increased mortality (Figure 3A, Table 3, and Appendix A). It should be noted that our results are in line with previous hormesis studies conducted by Ostap-Cheć et al. [59]. Those earlier studies showed that individuals exposed to alcohol concentrations of 0.5% and 1% did not exhibit hormesis, and most life-history parameters of bees exposed to these ethanol concentrations, including lifespan, did not differ from controls.

The toxic effect of ethanol was also observed in the group of workers infected with *Vairimorpha* spp. spores; however, in this case, bees exposed to high ethanol levels (5% and above) had significantly shorter lifespans compared to those in the control and lower ethanol concentration groups [43]. Additionally, our results reaffirmed that bees unexposed to *Vairimorpha* spp. spores generally lived longer, emphasizing the detrimental impact this microsporidium has on honeybee health [58,60].

One particularly intriguing finding of this study is that honeybees infected with *Vairimorpha* spp. spores and exposed to moderate ethanol concentrations (specifically 0.625% and 1.25%) exhibited the longest lifespans compared to those subjected to higher ethanol levels or no ethanol at all (Figure 3B). This unexpected result aligns with the hormetic model [1], which posits that low doses of certain substances can have protective or beneficial effects, whereas higher doses become toxic. In this context, it appears that low-level ethanol exposure may mitigate some of the detrimental impacts of *Vairimorpha* spp. infection. This dual nature of substance exposure, where low and high doses yield vastly different outcomes, is further illustrated by earlier reports indicating that, despite ethanol’s adverse effects, worker bees often preferred solutions with 1.25% or 2.5% ethanol over pure sugar solutions [39,42].

We also observed varying degrees of *Vairimorpha* spp. spore load across different experimental groups, revealing a significant association between ethanol dosage and spore concentration in honeybee digestive tracts. Bees exposed to higher ethanol concentrations (5% and 10%) exhibited lower *Vairimorpha* spp. spore counts compared to those exposed to lower concentrations (0%, 0.0313%, and 2.5%) (Figure 2). This suggests a potentially complex interaction where higher ethanol levels may inhibit *Vairimorpha* spp. proliferation, although the adverse effects of high ethanol concentrations on bee health must not be overlooked. More interesting is that bees exposed to 0.625% and 1.25% exhibited lower *Vairimorpha* spp. spore counts compared to both those from groups exposed to 0%, 0.0313 and 2.5% as well as exposed to 5% of ethanol. This duality of ethanol’s effects in this context becomes increasingly evident. While low doses appear beneficial for enhancing the lifespan of bees infected with *Vairimorpha* spp., high doses lead to impaired function and increased mortality. This is reminiscent of findings in other studies where low levels of toxins, like neonicotinoids [24] or heavy metals [9,11,14], resulted in hormetic responses that unexpectedly increased resilience and fitness in certain contexts [19,61]. However, careful consideration must be given to long-term effects, as the benefits gained from low ethanol exposure could be offset by potential trade-offs in reproductive success and overall colony health, particularly when considering that high ethanol concentrations can lead to cognitive and motor impairments.

We anticipated that alcohol would negatively affect *Vairimorpha* spp. spores in a concentration-dependent manner. However, our results indicate a more complex scenario. Bees exposed to ethanol concentrations of 2.5% and 5% had a higher number of *Vairimorpha* spp. spores than those exposed to lower concentrations of 0.625% and 1.25%. Similar findings were reported by Ptaszyńska et al. [43], where exposure to a 5% alcohol concentration resulted in a greater concentration of *Vairimorpha* spp. spores compared to both higher and lower concentrations. Additionally, a faster proliferation of *Vairimorpha* spp. and increased mortality among infected workers was observed following exposure to CO_2_ spores used for inoculation [62]. When dissolved in water, carbon dioxide creates carbonic acid, acidifying the spore solution [62], which likely contributes to the decline of the regenerative crypts within the midgut. This acidification may increase *Vairimorpha* spp. infection levels, impairing intestinal function, reducing nutrient absorption, and leading to malnutrition in bees, ultimately resulting in increased mortality rates [63].

The implications of our findings extend beyond individual bees to entire colonies. *Vairimorpha* spp. infections pose a considerable threat to honeybee populations, leading to decreased foraging efficiency, impaired immune functions, and increased vulnerability to other diseases [53,54,55,64]. The assessment of ethanol as a potential mitigating factor offers a novel perspective on managing Nosema infections. Beekeepers have long since employed various treatments to combat *Vairimorpha* spp., yet the efficacy of ethanol has been met with mixed results in other studies [43]. Our research complements these previous findings, framing ethanol as a substance that may harbor hormetic potentials, contingent on its concentration. Moreover, the study raises several pertinent questions regarding honeybee foraging behavior and the ecological interactions of ethanol in floral resources. The apparent preference of bees for sugar solutions containing low ethanol concentrations over pure sugar solutions [42,44] indicates a possible evolutionary adaptation that may not only assist in combating pathogens but may also enhance energetic resilience. Further exploration into the cues that drive this preference could provide insights into the evolutionary dynamics among plants, pollinators, and environmental stressors.

In conclusion, the exploration of ethanol hormesis in honeybees infected with *Vairimorpha* spp. opens a pathway for future research and management strategies. It is evident that the interactions between pollinators and their environment are intricate and multifaceted. The delicate balance between beneficial low-dose exposures and harmful high-dose consequences must be further scrutinized, especially as pollinators face mounting challenges from pathogens, pesticides, and environmental changes.

## Figures and Tables

**Figure 1 animals-15-03316-f001:**
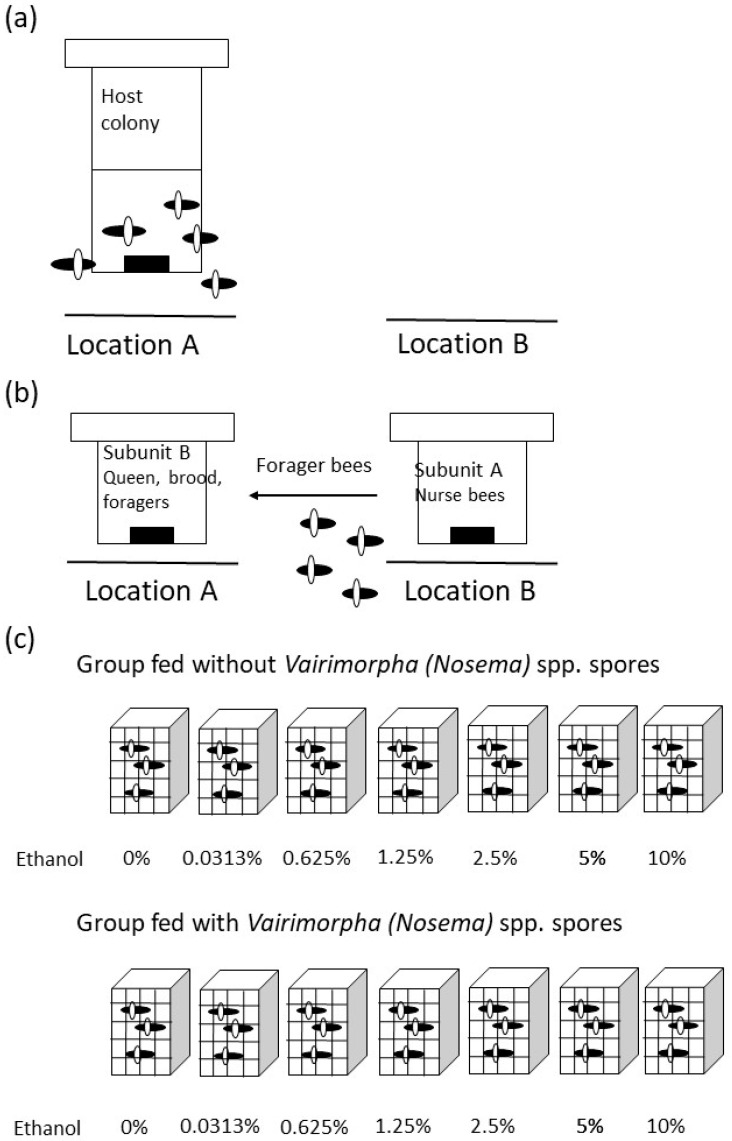
Scheme of the Experiment (presented for a single colony) (**a**) Each of the four original host colonies comprising nurse, bees, foragers, queens, larvae, and marked workers remained in Location 1. (**b**) All host colonies were divided into two parts to separate foragers from nurse bees. Subunit A, which included all adult workers along with combs containing honey and pollen, was relocated to a new site (Location 2) several meters away from the original hive site (Location 1). Subunit B consisting of the queen. empty combs. and combs with young larvae remained at the original location (Location 1). Consequently, foragers leaving Subunit A to gather food returned to Subunit B. (**c**) Groups of forager bees were captured and placed in cages in fourteen experimental groups—seven infected and seven uninfected with *Vairimorpha* (*Nosema*) spp.—and exposed to varying ethanol concentrations.

**Figure 2 animals-15-03316-f002:**
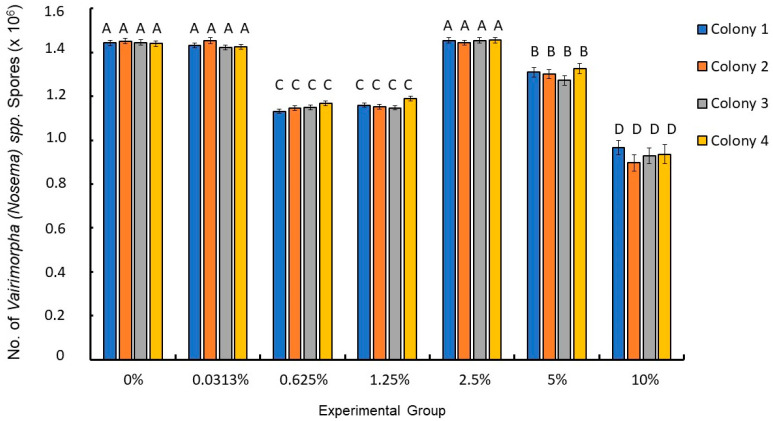
The number of *Vairimorpha* (*Nosema*) spp. spores in infected bees across different colonies and exposed to varying alcohol concentrations. The same letters indicate that there are no statistical differences between the bars. Bees from the uninfected *Vairimorpha* spp. groups were not included in the figure because the number of spores in these individuals was zero.

**Figure 3 animals-15-03316-f003:**
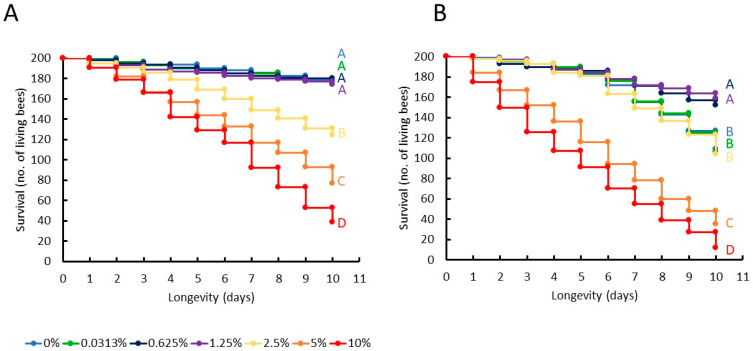
Survival curves for individuals from different experimental groups. (**A**) Not fed with *Vairimorpha* spp. spores, and (**B**) fed with *Vairimorpha* spp. spores. Data from bees of various colonies were combined. The same letters indicate no statistically significant differences between the curves.

**Table 1 animals-15-03316-t001:** The mean number of *Vairimorpha* spp. spores in experimental bees exposed to varying ethanol (EtOH) concentrations, including both infected and uninfected individuals. It also includes the standard error, the 95% confidence interval, and the sample size (N).

% EtOH	Vairimorpha Spores Feeding	Mean	SE	−95.00%	+95.00%	N
0%	No	0	3447.06	−6761	6761	177
0.0313%	No	0	3456.84	−6780	6780	176
0.625%	No	0	3447.06	−6761	6761	177
1.25%	No	0	3476.65	−6819	6819	174
2.5%	No	0	4118.36	−8078	8078	124
5%	No	0	5226.25	−10,251	10,251	77
10%	No	0	7439.50	−14,592	14,592	38
0%	Yes	1,445,064	4392.61	1,436,448	1,453,680	109
0.0313%	Yes	1,432,755	4454.33	1,424,018	1,441,492	106
0.625%	Yes	1,148,684	3719.75	1,141,388	1,155,980	152
1.25%	Yes	1,161,358	3636.95	1,154,225	1,168,492	159
2.5%	Yes	1,451,769	4496.96	1,442,949	1,460,590	104
5%	Yes	1,302,171	7751.78	1,286967	1,317,376	35
10%	Yes	934,000	13,238.69	908,033	959,967	12

**Table 2 animals-15-03316-t002:** The results of the log-rank test examining whether there are differences in lifespan among workers from different colonies. Tests were conducted for workers from different groups fed with *Vairimorpha* spp. and exposed to various concentrations of EtOH. The last row shows the analysis for all workers used in the experiment.

% EtOH	Vairimorpha Spores Feeding	χ2	N	Censored	Uncensored	df	*p*
0%	No	0.534	200	177	23	3	0.911
0.0313%	No	0.784	200	175	25	3	0.853
0.625%	No	0.539	200	177	23	3	0.910
1.25%	No	0.750	200	174	26	3	0.861
2.5%	No	1.540	200	125	75	3	0.673
5%	No	2.671	200	80	120	3	0.445
10%	No	0.759	200	39	161	3	0.859
0%	Yes	0.273	200	109	91	3	0.965
0.0313%	Yes	0.465	200	97	103	3	0.926
0.625%	Yes	0.135	200	152	48	3	0.987
1.25%	Yes	7.921	200	146	54	3	0.684
2.5%	Yes	0.684	200	104	96	3	0.877
5%	Yes	0.898	200	35	165	3	0.826
10%	Yes	0.888	200	12	188	3	0.828
Overall test		1.012	2800	1608	1192	3	0.798

**Table 3 animals-15-03316-t003:** The results of the log-rank test examined whether there are differences in lifespan among workers exposed to different ethanol (EtOH) concentrations. The analyses were conducted separately for groups that were not fed with *Vairimorpha* spp. spores and for those that were fed with spores.

Compared Groups	The Groups Fed Without *Vairimorpha* spp. Spores				The Groups Fed with *Vairimorpha* spp. Spores			
	Z	N	df	*p*	Z	N	df	*p*
0–0.0313%	−0.324	400	1	0.746	−0.445	400	1	0.656
0–0.625%	−0.031	400	1	0.975	4.225	400	1	<0.001
0–1.25%	−0.505	400	1	0.613	3.595	400	1	<0.001
0–2.5%	−6.031	400	1	<0.001	−0.586	400	1	0.558
0–5%	−10.068	400	1	<0.001	−8.885	400	1	<0.001
0–10%	−13.628	400	1	<0.001	−11.926	400	1	<0.001
0.0313–0.625%	0.293	400	1	0.769	4.700	400	1	<0.001
0.0313–1.25%	−0.185	400	1	0.853	4.067	400	1	<0.001
0.0313–2.5%	−2.236	400	1	0.025	−0.157	400	1	0.875
0.0313–5%	−9.772	400	1	<0.001	−8.867	400	1	<0.001
0.0313–10%	−13.341	400	1	<0.001	−11.779	400	1	<0.001
0.625–1.25%	−0.476	400	1	0.634	−0.650	400	1	0.516
0.625–2.5%	−5.954	400	1	<0.001	−4.742	400	1	<0.001
0.625–5%	−9.971	400	1	<0.001	−11.836	400	1	<0.001
0.625–10%	−13.501	400	1	<0.001	−14.377	400	1	<0.001
1.25–2.5%	−5.501	400	1	<0.001	−4.124	400	1	<0.001
1.25–5%	−9.548	400	1	<0.001	−11.047	400	1	<0.001
1.25–10%	−13.091	400	1	<0.001	−14.021	400	1	<0.001
2.5–5%	−4.505	400	1	<0.001	−8.378	400	1	<0.001
2.5–10%	−8.463	400	1	<0.001	−11.467	400	1	<0.001
5–10%	−4.055	400	1	<0.001	−3.657	400	1	<0.001

## Data Availability

All data generated or analyzed during this study are included in this published article (and its Appendix A files).

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
