# Peer review of "Ethanol Hormesis in Honeybees (*Apis mellifera* L.) Infected with *Vairimorpha (Nosema*) spp."

_animals, 2025, doi:10.3390/ani15223316_

Round 1

Reviewer 1 Report

Comments and Suggestions for Authors

This paper focuses on the specific field of "the role of ethanol hormesis in honeybees infected with Nosema sp, filling the gap in existing research. Although existing literature has separately reported the toxicity of ethanol to honeybees (e.g., high concentrations affecting motor ability) and the hazards of Nosema sp infection, it has not systematically explored the protective effect of low-concentration ethanol on infected honeybees. Through a controlled variable experiment (seven ethanol concentration gradients and two infection states), the paper for the first time confirms that "0.625% and 1.25% ethanol can simultaneously reduce Nosema spore load and extend the lifespan of infected honeybees", providing new evidence for the interaction mechanism between the two. However, the paper fails to conduct an in-depth analysis of the underlying mechanism. Although the "hormetic model" is mentioned in the discussion, it lacks support from molecular-level data (e.g., changes in gene expression and metabolites), which limits the depth of the content. In addition, there are errors in some data and formats, and it is recommended to publish the paper after revision.

  1. Lines 15-16: In the abstract, the statement "foraging bees were divided into groups of infected and uninfected individuals, and each group was exposed to varying ethanol concentrations" does not specify the sample size of each group (50 bees per group). Please add "each group (n=50)".
  2. Lines 21-23: The phrase "moderate ethanol levels might mitigate some harmful effects of Nosema sp. infections" needs to clarify that "moderate levels" refer to the experimentally confirmed "0.625% and 1.25%". Additionally, supplement information on "the percentage reduction in Nosema spore count compared to the control group and the number of days of lifespan extension at these concentrations".
  3. Line 147: Why was a spore suspension with a concentration of 1.75 × 10⁴ used to feed the honeybees?
  4. Line 153: The suitable temperature for rearing adult honeybees is usually 25-30°C with a humidity of 70%. Why were 36°C and a humidity of 50-60% adopted as the rearing conditions?
  5. Lines 207, 247 & 290: The "0.313%" here should be "0.0313%", which is a data entry error. It needs to be corrected to ensure the consistency of concentration gradients.
  6. Line 240: The "0.0625%" here should be "0.625%".
  7. "Nosema sp." and "Nosema spp." are used interchangeably in the paper. Please confirm the correct writing of the Latin scientific name.
  8. Figure 2: The grouping and significance labels are disorganized. Please present the differences between treatment groups across different colonies in separate subfigures.
  9. The Latin scientific names of species should be in italics. Please revise the entire paper accordingly.
  10. Foraging bees of different ages exhibit significant differences in physiological responses. The authors do not seem to have noticed this point. Please elaborate on the rationality of sample collection.
  11. The layout of the table content is chaotic. It is necessary to reorganize the column headings to ensure that each row of data corresponds to the correct group and that the numerical formats are unified.
  12. The clarity of the images throughout the paper is poor. Please provide high-resolution images again.

Author Response

Dear Editor

Dear Editor,

Thank you for the opportunity to resubmit our revised manuscript. We have addressed the reviewers’ comments and implemented the requested updates under the new title: “Ethanol hormesis in honeybees (Apis mellifera L.) infected with Vairimorpha (Nosema) spp.” The manuscript has been professionally language-edited to improve English. We have added a concise summary to enhance accessibility and included the following sections to meet journal requirements: Institutional Review Board Statement and Informed Consent Statement. Detailed responses to the reviewers’ queries are provided below.

Best regards,

Karolina Kuszewska

Regenerate

Reviewer #1

This paper focuses on the specific field of "the role of ethanol hormesis in honeybees infected with Nosema sp, filling the gap in existing research. Although existing literature has separately reported the toxicity of ethanol to honeybees (e.g., high concentrations affecting motor ability) and the hazards of Nosema sp infection, it has not systematically explored the protective effect of low-concentration ethanol on infected honeybees. Through a controlled variable experiment (seven ethanol concentration gradients and two infection states), the paper for the first time confirms that "0.625% and 1.25% ethanol can simultaneously reduce Nosema spore load and extend the lifespan of infected honeybees", providing new evidence for the interaction mechanism between the two. However, the paper fails to conduct an in-depth analysis of the underlying mechanism. Although the "hormetic model" is mentioned in the discussion, it lacks support from molecular-level data (e.g., changes in gene expression and metabolites), which limits the depth of the content. In addition, there are errors in some data and formats, and it is recommended to publish the paper after revision.

Thank you for the guidance. We acknowledge the lack of molecular-level data in this study and, instead, focus on phenotypic outcomes directly influenced by ethanol exposure in Nosema-infected honeybees; all individuals were inoculated with the same, uniform spore load to ensure comparability, and we measured lifespan along with spore counts to provide functional endpoints. Our data show that lifespan was longer in infected individuals exposed to ethanol at 0.625% and 1.25% compared with controls, which is consistent with a hormetic response, and the uniform spore exposure across groups supports this interpretation within the limits of a phenotypic study. We recognize that molecular analyses would strengthen mechanistic understanding; however, no further molecular work is planned at this time. We also apologize for formatting and typographical errors in earlier versions and have thoroughly revised the manuscript to improve language clarity and formatting in this current version; we hope these issues are no longer present.

  1. Lines 15-16: In the abstract, the statement "foraging bees were divided into groups of infected and uninfected individuals, and each group was exposed to varying ethanol concentrations" does not specify the sample size of each group (50 bees per group). Please add "each group (n=50)".

Thank you. We have added this information to the text.

  1. Lines 21-23: The phrase "moderate ethanol levels might mitigate some harmful effects of Nosema sp. infections" needs to clarify that "moderate levels" refer to the experimentally confirmed "0.625% and 1.25%". Additionally, supplement information on "the percentage reduction in Nosema spore count compared to the control group and the number of days of lifespan extension at these concentrations".

Thank you for the comment. We have revised this section in the manuscript to present concrete values (line 46-49 in new version of MS), and we have added to the supplementary data a table summarizing the mean lifespans of workers in each group and the mean changes in Vairimorpha spp. (Nosema) spore counts relative to the control.

  1. Line 147: Why was a spore suspension with a concentration of 1.75 × 10⁴ used to feed the honeybees?

That spore concentration was used due to prior pilot studies and earlier paper (Kuszewska K, Żmuda A, Gajda AM. 2025 Age and attitude: How longevity influences cognitive biases in honeybee workers. Proc. R. Soc. B 292: 20251696). This spore concentration leads to the appearance of disease symptoms after a few days, while the infection does not progress very quickly, allowing infected bees to remain alive for some time.

  1. Line 153: The suitable temperature for rearing adult honeybees is usually 25-30°C with a humidity of 70%. Why were 36°C and a humidity of 50-60% adopted as the rearing conditions?

We conducted our experiments considering the temperature inside the bee nest, which is in the range of 33–36 °C, and a relative humidity of about 52% (e.g. Abou-Shaara, H.F., Owayss, A.A., Ibrahim, Y.Y. et al. A review of impacts of temperature and relative humidity on various activities of honey bees. Insect. Soc. 64, 455–463 (2017). https://doi.org/10.1007/s00040-017-0573-8; Tautz J, Maier S, Groh C, Roessler W, Brockmann A (2003) Behavioral performance in adult honey bees is influenced by the temperature experienced during their pupal development. Proc Nat Acad Sci USA 100:7343–7347). We concluded that nest-like conditions are most optimal for all workers and generate the least stress, as we aimed to exclude potential stressful conditions. This is a common condition used in many studies (e.g. Geoffrey R Williams, Cédric Alaux, Cecilia Costa, Támas Csáki, Vincent Doublet, Dorothea Eisenhardt, Ingemar Fries, Rolf Kuhn, Dino P McMahon, Piotr Medrzycki, Tomás E Murray, Myrsini E Natsopoulou, Peter Neumann, Randy Oliver, Robert J Paxton, Stephen F Pernal, Dave Shutler, Gina Tanner, Jozef J M van der Steen & Robert Brodschneider (2013) Standard methods for maintaining adult Apis mellifera in cages under in vitro laboratory conditions, Journal of Apicultural Research, 52:1, 1-36, DOI: 10.3896/IBRA.1.52.1.04;  Kuszewska K, Żmuda A, Gajda AM. 2025 Age and attitude: How longevity influences cognitive biases in honeybee workers. Proc. R. Soc. B 292: 20251696)

  1. Lines 207, 247 & 290: The "0.313%" here should be "0.0313%", which is a data entry error. It needs to be corrected to ensure the consistency of concentration gradients.

Thank you; we have corrected this error in the new version of the manuscript.

  1. Line 240: The "0.0625%" here should be "0.625%".

Thank you; we have corrected this error in the new version of the manuscript.

  1. "Nosema sp." and "Nosema spp." are used interchangeably in the paper. Please confirm the correct writing of the Latin scientific name.

Thank you; we have changed the name from Nosema sp. to Vairimorpha spp., adding in parentheses the name Nosema. We did this due to the recent nomenclature change as well as the suggestion of the second reviewer.

  1. Figure 2: The grouping and significance labels are disorganized. Please present the differences between treatment groups across different colonies in separate subfigures.

Thank you for this comment. However, in this case we respectfully disagree with the reviewer’s opinion. In the figure, colonies are clearly distinguishable by contrasting colors, and the ethanol doses are easy to identify as clusters of four bars separated by gaps and labeled on the X-axis. The letters above each bar indicate where differences exist (different letters) or do not (same letters), as noted in the figure caption. This layout allows comparisons both within each colony and between colonies under the same or different ethanol exposures. Creating separate plots for each colony would hinder cross-colony comparisons. That said, if the Editor prefers a figure composed of four separate panels, we will be happy to prepare it.

  1. The Latin scientific names of species should be in italics. Please revise the entire paper accordingly.

Thank you; we have corrected this error in the new version of the manuscript.

  1. Foraging bees of different ages exhibit significant differences in physiological responses. The authors do not seem to have noticed this point. Please elaborate on the rationality of sample collection.

Of course, variation in the ages of the worker foragers could have influenced the results—specifically, by increasing variance (since bees were randomly assigned to groups, each could contain similar numbers of younger and older individuals), thereby reducing the likelihood of detecting statistical differences. The fact that we still observed differences indicates that the treatment effect was strong. Moreover, even if we had marked bees and used individuals of the same chronological age, their social status (i.e., task) depends not only on age but also on expected lifespan. Consequently, one 20-day-old bee might still be working inside the nest while another has been foraging for several days—implying distinct physiologies (e.g.Woyciechowski, M., Moroń, D. Life expectancy and onset of foraging in the honeybee (Apis mellifera). Insect. Soc. 56, 193–201 (2009). https://doi.org/10.1007/s00040-009-0012-6.

In our study, we needed to obtain 700 foragers from each colony (7 ethanol treatments × 2 Vairimorpha [Nosema] spp. groups × 50 individuals), which would not have been feasible from a single age cohort. Although it is possible to rear around 2,000 workers of the same age, by the time all or most would begin foraging, a substantial proportion would likely have died. Therefore, accepting the risk of reduced power, we opted to collect foragers of different ages directly from the hive.

  1. The layout of the table content is chaotic. It is necessary to reorganize the column headings to ensure that each row of data corresponds to the correct group and that the numerical formats are unified.

Thank you—we’ve improved the table formatting.

  1. The clarity of the images throughout the paper is poor. Please provide high-resolution images again.

Thank you, we’ve uploaded higher-quality images.

Reviewer 2 Report

Comments and Suggestions for Authors

The phenomenon of hormesis cannot be denied, and a close-up examination of biological effects of substances on living organisms should always consider this possibility. Honey bee is a key pollinator and important industrial insect for honey production. There is a broad spectrum of diseases, including viruses, bacteria, protists, fungi, and mites. The Nosema disease, caused by Vairimorpha ceranae, V. apis or V. neumanni, is a widespread problem in apiaries around the world. Numerous studies aim at mitigating its harm to apiculture. The use of substances which act as disinfectants but also may positively impact honey bee health is a promising area of research. This makes the manuscript actual and interesting for the audience of the journal, perfectly fitting its aims and scope.

The manuscript is well written and easy to read. The Introduction covers the topic sufficiently. Methods are described meticulously, without unnecessary details. Experimental design is sound.

It can be recommended for publication after some correction accordign to the comments below.

The systematics of the causative agent of Nosema disease should be re-considered. Curiously, “Nosema” means “disease” in Greek and it’s okay to use it as a conventional disease name, yet the taxonomic position of the microsporidia which cause these infections is different, they all belong to the genus of Vairimorpha. This taxonomic decision was first proposed in 2020, based upon a molecular phylogenetic study (https://doi.org/10.1016/j.jip.2019.107279), further argued in 2024 due to some formal reasons, such as a limited number of analyzed loci (https://doi.org/10.1016/j.jip.2024.108146), but robustly confirmed in 2025 with the help of whole-genome sequencing and phylogenomics approaches (https://doi.org/10.1016/j.jip.2025.108376). Little evidence supports the idea that the microsporidia infecting honey bees should be attributed to the genus of Nosema.

The use of italics for Latin epithets of genus and species should be checked throughout.

Lines 50-52: The effects of hormesis in different representatives of insects should be disclosed in a greater detail, including symbiotic systems

Line 56: this should be explained what is the nature of substances on which these commercial names are based – insecticides, herbicides etc. Are there studies involving other synthetic substances in honey bees showing hormesis?

Lines 76-78: it would be nice to clarify under which conditions bees can be exposed to high concentrations of ethanol. Does it have repellent properties, protecting bees from consumption?

Lines 92-94: please be careful about such conclusions. First, “initially” the “Nosema ceranae” infection was identified in the Eastern honey bee Apis ceranae not “in hot and humid climates”, but in a particular country. The reference number 44 describes identification of this species not in A. ceranae but in A. mellifera. The fact that this microsporidium was found in other countries later doesn’t mean that it has spread there after first discovery. For example, you can read the work by Ivernizzi et al. (2009) which proved that “Nosema ceranae” was present in Uruguay before 1990 (https://www.sciencedirect.com/science/article/pii/S0022201109000585).

Line 95: the enumeration of countries is arbitrary, why you have chosen these examples only? There are plenty of other countries with consistent reports of “Nosema ceranae”, including Russia, Canada, Brazil, Argentina, Japan, Australia etc., etc. Besides, by all types of geographic and political categorization of countries, Poland belongs to Europe and this is strange to see it separately from Europe in this enumeration

Line 140: check use of dot

Line 240 vs 245-246: I find a controversy here, first you write “low ethanol doses of 0.0313%, 0.0625%, and 1.25% did not impact lifespan” but then “the longest-lived bees were those exposed to 0.625% and 1.25% ethanol”. Please be consistent with this and always indicate whether the differences were statistically significant

Line 257: check the use of dot

Lines 261-264: this style of narrative (particular values, reference to tables and figures) is suitable for Results, where this information is already given. Discussion should not repeat the Result section but rather provide a summary in a more generalized form with appropriate conclusions.

Lines 264-265 and 272-274: as similar results have been obtained by previous study, the scientific novelty of the present work should be highlighted

Line 290: see comments for Lines 261-264

Figures should be provided in the Results section

Figures 2 & 3: which statistical approach was exploited for survival comparisons?

Author Response

Dear Editor

Dear Editor,

Thank you for the opportunity to resubmit our revised manuscript. We have addressed the reviewers’ comments and implemented the requested updates under the new title: “Ethanol hormesis in honeybees (Apis mellifera L.) infected with Vairimorpha (Nosema) spp.” The manuscript has been professionally language-edited to improve English. We have added a concise summary to enhance accessibility and included the following sections to meet journal requirements: Institutional Review Board Statement and Informed Consent Statement. Detailed responses to the reviewers’ queries are provided below.

Best regards,

Karolina Kuszewska

Regenerate

 Reviewer #2

The phenomenon of hormesis cannot be denied, and a close-up examination of biological effects of substances on living organisms should always consider this possibility. Honey bee is a key pollinator and important industrial insect for honey production. There is a broad spectrum of diseases, including viruses, bacteria, protists, fungi, and mites. The Nosema disease, caused by Vairimorpha ceranae, V. apis or V. neumanni, is a widespread problem in apiaries around the world. Numerous studies aim at mitigating its harm to apiculture. The use of substances which act as disinfectants but also may positively impact honey bee health is a promising area of research. This makes the manuscript actual and interesting for the audience of the journal, perfectly fitting its aims and scope.

The manuscript is well written and easy to read. The Introduction covers the topic sufficiently. Methods are described meticulously, without unnecessary details. Experimental design is sound.

It can be recommended for publication after some correction accordign to the comments below.

The systematics of the causative agent of Nosema disease should be re-considered. Curiously, “Nosema” means “disease” in Greek and it’s okay to use it as a conventional disease name, yet the taxonomic position of the microsporidia which cause these infections is different, they all belong to the genus of Vairimorpha. This taxonomic decision was first proposed in 2020, based upon a molecular phylogenetic study (https://doi.org/10.1016/j.jip.2019.107279), further argued in 2024 due to some formal reasons, such as a limited number of analyzed loci (https://doi.org/10.1016/j.jip.2024.108146), but robustly confirmed in 2025 with the help of whole-genome sequencing and phylogenomics approaches (https://doi.org/10.1016/j.jip.2025.108376). Little evidence supports the idea that the microsporidia infecting honey bees should be attributed to the genus of Nosema.

Thank you for your comment. We have updated the manuscript by changing the taxon name from Nosema sp. to Vairimorpha sp., and we added the necessary citations. Consequently, the title has been revised to reflect this change. We also modified the introduction section by including information about the taxonomic name change

The use of italics for Latin epithets of genus and species should be checked throughout.

Thank you; we hope that all words are now properly formatted.

Lines 50-52: The effects of hormesis in different representatives of insects should be disclosed in a greater detail, including symbiotic systems

Thank you for this comment. In the initial version of the manuscript we briefly described the progression from hormesis at the general organismal level (across many organisms) through invertebrates, and then we moved on to Hymenoptera. We agree that the topic is much broader, not only as a phenomenon in itself but also within insects, including their symbiotic systems. However, we believe that such a detailed description in the introduction of other insects and systems beyond those we studied would substantially lengthen the manuscript. This is not a review article, where such information would typically be included. If the Editor requires a more detailed introduction, we are prepared to undertake this challenge.

Line 56: this should be explained what is the nature of substances on which these commercial names are based – insecticides, herbicides etc. Are there studies involving other synthetic substances in honey bees showing hormesis?

Thank you for this comment. We have added this information to the new version of the manuscript.

Lines 76-78: it would be nice to clarify under which conditions bees can be exposed to high concentrations of ethanol. Does it have repellent properties, protecting bees from consumption?

We added information about natural exposition to ethanol. Honey bees find the taste of ethanol aversive in water, but adding sucrose masks this aversive taste even in higher concetration (Mustard, J.A., Oquita, R., Garza, P. and Stoker, A. (2019), Honey Bees (Apis mellifera) Show a Preference for the Consumption of Ethanol. Alcohol Clin Exp Re, 43: 26-35. https://doi.org/10.1111/acer.13908).

Lines 92-94: please be careful about such conclusions. First, “initially” the “Nosema ceranae” infection was identified in the Eastern honey bee Apis ceranae not “in hot and humid climates”, but in a particular country. The reference number 44 describes identification of this species not in A. ceranae but in A. mellifera. The fact that this microsporidium was found in other countries later doesn’t mean that it has spread there after first discovery. For example, you can read the work by Ivernizzi et al. (2009) which proved that “Nosema ceranae” was present in Uruguay before 1990 (https://www.sciencedirect.com/science/article/pii/S0022201109000585).

Thank you for this comment; we have changed these parts of the paragraphs (line 119-130 in new MS)

Line 95: the enumeration of countries is arbitrary, why you have chosen these examples only? There are plenty of other countries with consistent reports of “Nosema ceranae”, including Russia, Canada, Brazil, Argentina, Japan, Australia etc., etc. Besides, by all types of geographic and political categorization of countries, Poland belongs to Europe and this is strange to see it separately from Europe in this enumeration

Thank you for this comment; we have changed these parts of the paragraphs (line 131-141in new MS)

Line 140: check use of dot

Thank you for this comment; there were clearly too many dots in inappropriate places, so we removed them, and we hope that everything is now in order

Line 240 vs 245-246: I find a controversy here, first you write “low ethanol doses of 0.0313%, 0.0625%, and 1.25% did not impact lifespan” but then “the longest-lived bees were those exposed to 0.625% and 1.25% ethanol”. Please be consistent with this and always indicate whether the differences were statistically significant

Thank you for that remark. The two sentences referred to different worker groups: the first (“low ethanol doses of 0.0313%, 0.0625%, and 1.25% did not impact lifespan” ) concerned workers not fed with spores of Vairimorpha spp. (Nosema), while the second (“the longest-lived bees were those exposed to 0.625% and 1.25% ethanol”) referred to workers that were fed spores. However, since we agree that this was written unclearly, we have revised this fragment of the text in our new manuscript and hope that it will no longer mislead readers.

Line 257: check the use of dot

Thank you for this comment. We have corrected these parts

Lines 261-264: this style of narrative (particular values, reference to tables and figures) is suitable for Results, where this information is already given. Discussion should not repeat the Result section but rather provide a summary in a more generalized form with appropriate conclusions.

We agree with the view that the discussion should not repeat results; however, in this case we repeat only a small portion so that readers can quickly reference which part is under discussion. In our view, this facilitates reading. Nevertheless, if the reviewers and the editor determine that this brief repetition should be removed, we will do so.

Lines 264-265 and 272-274: as similar results have been obtained by previous study, the scientific novelty of the present work should be highlighted

We agree that the result is similar, which we also discuss; we do this in order to show that our data reflect what has been reported in other studies, and that, in fact, similar methods yield similar results. However, in the next paragraph we discuss what is more interesting to us and, of course, new to science.

Line 290: see comments for Lines 261-264

Similarly as above we agree with the view that the discussion should not repeat results; however, in this case we repeat only a small portion so that readers can quickly reference which part is under discussion. In our view, this facilitates reading. Nevertheless, if the reviewers and the editor determine that this brief repetition should be removed, we will do so.

Figures should be provided in the Results section

Thanks, we have changed this in the new version of MS.

Figures 2 & 3: which statistical approach was exploited for survival comparisons?

As described in the Methods (line 211-220 in the MS), to assess differences in the number of Vairimorpha spp. spores among groups, a three-way ANOVA was performed, followed by post-hoc Tukey tests for relevant factors across different sample sizes (N).

 Regarding Figures 3a and 3b, which illustrate the survival of individuals from non-fed (3a) and fed (3b) groups, Vairimorpha spp. analyses of survival were conducted separately by comparing multiple curves (log-rank test, incorporating data from individuals that survived the experiment—i.e., up to day 10). If significant differences were found, multiple pairwise comparisons for these curves were also undertaken using the log-rank test. The results for Figure 3 are also reported in Tables 1–3, and all other data are provided in the supplementary data (which includes the original data that can be re-tested). In the figure legends we did not include statistical data, as we did not want to repeat it in such a brief text.

Round 2

Reviewer 2 Report

Comments and Suggestions for Authors

The paper is sufficiently improved. All questionable part are rewritten. All reviewer's comments are responded and text corrected accordingly. Some technical remarks require attention, as listed below

Vairimorpha (Nosema) spp. is a correct form of introduction of synonymy at first mentioning in title, abstract,  figure legends, and main text. However, after first mentioning, it becomes redundant

Line 16 and further: double dot should be changed to single one

Lines 19-20: details of experimental design are not necessary in Simple Summary

Line 22: “findings” do not “show”, consider rewriting

Line 27 and further: “interactions between ethanol, infection, and bee health” – “interactions” is used when the components influence each other. I doubt that infection (or bee health) brings any effect to ethanol

Line 80: the details given in parentheses for different substances are heterogenous: in case of insecticide you refer to the active compound (Clothianidin) and decipher its chemical class (neonocitinoid), while for herbicide, you provide commercial product name (Roundup) and explain that herbicide is a weed killer, without referencing to its chemical class (and without mentioning that insecticide is an insect killer). Please be consistent with definitions.

Line 100: ofinsects – space missing

Lines 107-108: These examples show how species may benefit from ethanol stress – which examples, which species, where the benefits are shown?

Line 116: a dot a space at the end of the sentence missing

Line 120 and elsewhere: “spp” doesn’t require italics

Line 123: “genera” is plural, “genus” is singular

Lines 125-126: the explanation given in parentheses is misleading, as “Nosematidae” was described later than “Nosema” and “nosemosis” were introduced. It can be completely removed or rewritten

Line 133: odd punctuation

Line 404: odd punctuation

Line 406: what is “energetic resilience”?

Lines 435-437 – I believe that simple “Not applicable” is sufficient in this section

Lines 439-440: the construction of the phrase “in order to language correction” clearly depicts that AI model exploited is far from perfection to be used for language correction

Author Response

The paper is sufficiently improved. All questionable part are rewritten. All reviewer's comments are responded and text corrected accordingly. Some technical remarks require attention, as listed below

Vairimorpha (Nosema) spp. is a correct form of introduction of synonymy at first mentioning in title, abstract,  figure legends, and main text. However, after first mentioning, it becomes redundant

Thank you; we have removed the synonym from the later part of the manuscript.

Line 16 and further: double dot should be changed to single one

Thank you for your note; we have removed the unnecessary dot

Lines 19-20: details of experimental design are not necessary in Simple Summary

Thank you for the note. We will rewrite a simple summary.

Line 22: “findings” do not “show”, consider rewriting

Thank you for the note. We will rewrite a simple summary.

Line 27 and further: “interactions between ethanol, infection, and bee health” – “interactions” is used when the components influence each other. I doubt that infection (or bee health) brings any effect to ethanol

Thank you for your note. We will revise the summary to remove the discussion of interaction and replace it with the term "associations”.

Line 80: the details given in parentheses for different substances are heterogenous: in case of insecticide you refer to the active compound (Clothianidin) and decipher its chemical class (neonocitinoid), while for herbicide, you provide commercial product name (Roundup) and explain that herbicide is a weed killer, without referencing to its chemical class (and without mentioning that insecticide is an insect killer). Please be consistent with definitions.

Thank you. We have unified this part of the manuscript—instead of the commercial name Roundup, we have given the active component the name Glyphosate and clarified its chemical class as phosphonate.

Line 100: ofinsects – space missing

Thank you; we have separated these two words.

Lines 107-108: These examples show how species may benefit from ethanol stress – which examples, which species, where the benefits are shown?

We agree that this sentence made no sense in our paper, and therefore we decided to remove it.

Line 116: a dot a space at the end of the sentence missing

Thank you; we added a dot.

Line 120 and elsewhere: “spp” doesn’t require italics

Thank you; we changed this throughout the manuscript

Line 123: “genera” is plural, “genus” is singular

Thank you; we have revised this.

Lines 125-126: the explanation given in parentheses is misleading, as “Nosematidae” was described later than “Nosema” and “nosemosis” were introduced. It can be completely removed or rewritten

Thank you for the note. We have removed this fragment from the text

Line 133: odd punctuation

Thank you; we have revised this.

Line 404: odd punctuation

Thank you; we have revised this.

Line 406: what is “energetic resilience”?

Thank you for your input. We have removed that fragment from the manuscript.

Lines 435-437 – I believe that simple “Not applicable” is sufficient in this section

Thank you; we have revised this.

Lines 439-440: the construction of the phrase “in order to language correction” clearly depicts that AI model exploited is far from perfection to be used for language correction

Thank you; of course, this sentence construction was not correct, and we have changed it.